# C-MCTS:
# SAFE PLANNING WITH MONTE CARLO TREE SEARCH

## ABSTRACT

The Constrained Markov Decision Process (CMDP) formulation allows to solve safety-critical decision making tasks that are subject to constraints. While CMDPs have been extensively studied in the Reinforcement Learning literature, little attention has been given to sampling-based planning algorithms such as MCTS for solving them. Previous approaches perform conservatively with respect to costs as they avoid constraint violations by using Monte Carlo cost estimates that suffer from high variance. We propose Constrained MCTS (C-MCTS), which estimates cost using a safety critic that is trained with Temporal Difference learning in an offline phase prior to agent deployment. The critic limits exploration by pruning unsafe trajectories within MCTS during deployment. C-MCTS satisfies cost constraints but operates closer to the constraint boundary, achieving higher rewards than previous work. As a nice byproduct, the planner is more efficient w.r.t. planning steps. Most importantly, under model mismatch between the planner and the real world, C-MCTS is less susceptible to cost violations than previous work.

## 1 INTRODUCTION

Monte Carlo Tree Search (MCTS) is a decision-making algorithm that employs Monte Carlo methods across the decision space, evaluates their outcome with respect to a given reward/objective, and constructs a search tree focusing on the most promising sequences of decisions (Browne et al., 2012; Świechowski et al., 2022). The success of MCTS can be attributed to the asymmetry of the trees constructed, which ensures better exploration of promising parts of the search space. Also, the possibility of using neural networks as heuristics to guide the search tree has helped tackle complex and high-dimensional problems with large state and action spaces (Schrittwieser et al., 2020a).

In spite of its successful application in several diverse domains, the standard, single-objective MCTS algorithm is unsuitable for a large class of real-world problems that apart from optimizing an objective function, also require a set of constraints to be fulfilled. These types of problems are usually modeled as Constrained Markov Decision Processes (CMDPs) (Altman, 1999) and specialized algorithms are used to solve the underlying constrained optimization problem.

Typical examples of such algorithms include approaches that rely on an expert knowledge base to create a safe action set (Hoel et al., 2020; Mohammadhasani et al., 2021; Mirchevska et al., 2018), Lagrangian relaxation methods that update primal and dual variables incrementally online and learn safe policies (Ding et al., 2020; Paternain et al., 2019), approaches that learn separate reward and cost/constraint signals to train a safe-aware policy both in Markov Decision Process (MDP) (Bharadhwaj et al., 2020; Srinivasan et al., 2020; Yang et al., 2022) and Robust Markov Decision Process (RMDP) environments (Tamar et al., 2014; Mankowitz et al., 2020), as well as methods that utilize uncertainty-aware estimators like Gaussian Processes to balance the exploration/exploitation risk (Wachi et al., 2018; Hewing et al., 2019). Finally, a notably different way is to model problems with constraints using temporal logic specifications (Demri & Gastin, 2012), and incorporating them as soft constraints to solve a CMDP (Guo & Zavlanos, 2018; Kalagarla et al., 2022).

We propose Constrained MCTS (C-MCTS)[1], a novel MCTS-based approach for solving Constrained Markov Decision Process (CMDP) problems, see Fig. 1. We utilize a high-fidelity simulator to collect different sets of trajectories under different safety constraint satisfaction levels. Utilizing a simulator has several benefits, since violating cost-constraints has no real-world implications. Also, we can

---

[1]Our implementation will be publicly available after acceptance. Please find it in the supplementary material

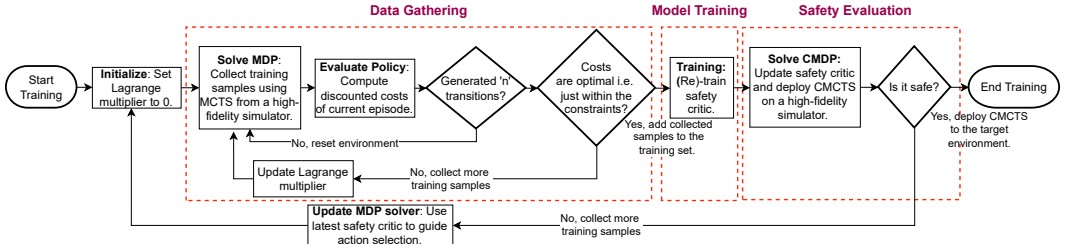

Figure 1: Simplified flow of training phase in C-MCTS.

construct scenarios with high safety implications that have rare real world occurrences. The samples collected are used to train a safety critic offline, which is used during deployment within MCTS to make cost predictions and avoid tree expansion to unsafe states. The proposed method manages to construct a deeper search tree with fewer planning iterations compared to the state of the art, while discovering solutions that operate safely closer to the cost-constraint, thus leading to higher rewards.

## 2 CONSTRAINED MARKOV DECISION PROCESSES

### 2.1 METHODS WITH LAGRANGE MULTIPLIERS

A CMDP can be defined by the tuple $\langle S, A, P, R, \mathbf{C}, \hat{\mathbf{c}}, \gamma, s_0 \rangle$ where $S$ is the set of states $s$, $A$ is the set of actions $a$, $P$ defines the probability of transitioning from $s \in S$ to $s' \in S$ for action $a \in A$ executed at $s$, $R$ is a reward function that returns a one-step reward for a given action $a$ at a state $s$, $\gamma \in [0, 1)$ is the discount factor, and $s_0 \in S$ is the initial state distribution. Following the notation convention of Lee et al. (2018), $\mathbf{C} = \{C_m\}_{1\ldots M}$ is a set of $M$ non-negative cost functions, with $\hat{\mathbf{c}} = \{c_m\}_{1\ldots M} \in [0, 1]$ their respective thresholds. For the remainder of the text, we assume only one constraint function $C$ with its respective threshold $\hat{c}$, to simplify the notation. The optimal policy $\pi^* \in \Pi$ in such a framework is a policy that belongs to a (parametric) policy class $\Pi$ that maximizes the expected discounted cumulative reward $V_R^\pi(s_0)$, while satisfying all the constraints on the expected discounted cumulative cost $V_C^\pi(s_0)$, as follows:

$$\max_{\pi \in \Pi} V_R^\pi(s_0) = \mathbb{E}_\pi \left[ \sum_{t=0}^\infty \gamma^t R(s_t, a_t)|s_0 \right] \quad s.t. \quad V_C^\pi(s_0) = \mathbb{E}_\pi \left[ \sum_{t=0}^\infty \gamma^t C(s_t, a_t)|s_0 \right] \leq \hat{c}. \quad (1)$$

Note that depending on the context we will use the definitions of (1) or the notion of the state-action expected discounted cumulative reward/cost (also known as the state-action value function), defined (for cost) as follows:

$$Q_C^\pi(s, \pi(s)) = \mathbb{E}_\pi \left[ \sum_{t=0}^\infty \gamma^t C(s_t, a_t)|s_0 = s \right] \triangleq V_C^\pi(s|s_0 = s). \quad (2)$$

Similar to assumptions of previous work (see e.g., Tessler et al. (2018) and the robust constraint objective (Eq. 2) from Mankowitz et al. (2020)), we prioritize the constraint satisfaction part of (1).

**Definition 1.** *(Tessler et al., 2018) A feasible solution of the constrained optimization problem defined in (1) is a solution that satisfies $V_C^\pi(s_0) \leq \hat{c}$.*

One approach to address the problem in (1) is using the Lagrange multiplier technique (see Bertsekas (2014)), which transforms the posed problem into an unconstrained one:

$$\min_{\lambda \geq 0} \max_{\pi \in \Pi} L(\lambda, \pi) = \min_{\lambda \geq 0} \max_{\pi \in \Pi} \left[ V_R^\pi(s_0) - \lambda \left( V_C^\pi(s_0) - \hat{c} \right) \right]. \quad (3)$$

For solving the Lagrangian, we can define the following:

**Definition 2.** *(Tessler et al., 2018) The penalized reward function is defined as $r(\lambda, s, a) = r(s, a) - \lambda c(s, a)$. The penalized expected discounted cumulative reward function is defined as $V_R^\pi(\lambda, s) = V_R^\pi(s) - \lambda V_C^\pi(s)$.*

Many approaches (e.g., Tessler et al. (2018); Ding et al. (2020)) parameterize the policy with parameters $\theta$ (e.g., a neural network policy) and directly apply consecutive steps of policy optimization

(e.g., using actor-critic algorithms) and adaptations of the value of $\lambda$, until the solution converges to a policy that respects the constraints. Others (e.g., Srinivasan et al. (2020); Yang et al. (2022)), learn a "safety" critic separately and try to maximize the expected reward without violating the constraints.

## 2.2 MONTE CARLO TREE SEARCH FOR CMDPS

MCTS is a decision-making algorithm that can search large combinatorial spaces represented by trees. The search tree consists of nodes representing each state uniquely, and edges representing actions that connect these nodes. The algorithm is used iteratively to explore the state space and build statistical evidence about different decision paths. Based on the gathered statistics, an optimal set of actions is taken such that the expected cumulative reward is maximized. Each iteration of MCTS consists of four phases: (i) selection, (ii) expansion, (iii) simulation, and (iv) backpropagation.

MCTS for *discrete-action* CMDPs has only been little explored. To our knowledge, apart from the seminal work of Lee et al. (2018), previous work extended MCTS to multi-objective variants (Hayes et al., 2023) that attempt to construct local (Chen & Liu, 2019) or global (Wang & Sebag, 2012) Pareto fronts and determine the Pareto-optimal solution. These approaches report good results at the expense of higher computational costs, due to the need to compute a set of Pareto-optimal solutions. Lee et al. (2018) proposed Cost-Constrained Partially Observable Monte Carlo Planning, an MCTS algorithm to solve Constrained Partially Observable Markov Decision Process problems, which can be used to solve CMDP settings also – we will refer to this variation of the algorithm as Cost-Constrained Monte Carlo Planning (CC-MCP). CC-MCP uses a Lagrange formulation and updates the Lagrange multiplier while constructing the search tree based on accumulated cost statistics. The CMDP problem is formulated as an Linear Program (LP), and then the dual formulation is solved:

$$\min_{\lambda \geq 0} [V_R^*(\lambda, s) + \lambda \hat{c}] \tag{4}$$

Here, $V_R^*(\lambda, s)$ is the optimal penalized expected discounted cumulative reward function, and $\hat{c}$ are the cost constraints. As the objective function in Eq. 4 is piecewise-linear and convex over $\lambda$ (Lee et al., 2018), $\lambda$ can be updated using the gradient information $V_C^* - \hat{c}$, where $V_C^*$ are the costs incurred for an optimal policy with a fixed $\lambda$. Hence, the CMDP can be solved by iterating the following three steps: (i) Solve MDP with a penalized reward function (see Definition 2), (ii) evaluate $V_C^*$ for this policy, and (iii) update $\lambda$ using the gradient information. Steps (i) and (ii) can also be interleaved at a finer granularity, and this is the idea behind CC-MCP, where $\lambda$ is updated at every MCTS iteration based on the Monte Carlo cost estimate $\hat{V}_C$ at the root node of the search tree.

## 3 METHOD

CC-MCP has two major shortcomings: (1) it requires a large number of planning iterations for the tuning of the Lagrange multiplier (as it is tuned online and thus explores both unsafe and safe trajectories in the search tree), and (2) the performance is sub-optimal w.r.t costs, i.e., the agent acts conservatively. Moreover, the algorithm also relies on the planning model to calculate cost estimates, making it error-prone to use approximate planning models for fast planning at deployment.

In our approach, the training phase (see Fig. 1) consists of approximating a safety critic that is utilized by the MCTS policy during the deployment phase (without a Lagrange multiplier) for pruning unsafe trajectories/sub-trees. We define two simulators: a low-fidelity simulator, which is not accurate, but its low complexity allows for utilization in the *online* planning/rollout phase of the MCTS algorithm; and a high-fidelity one, utilized for data collection and evaluation of the safety critic training.

## 3.1 SAFETY CRITIC TRAINING

Instead of tuning the Lagrange multiplier online, C-MCTS varies this parameter in a pre-training phase (in simulation) to obtain different sets of trajectories with different safety levels ("data gathering" in Fig. 1). Here, following the standard training process in Lagrangian relaxation/augmentation settings (Bertsekas, 2014), training iterates between calculating a new value $\lambda_k$ in each $k$-th iteration of the data gathering loop and solving the $k$-th MDP (using MCTS) with the penalized reward function $r(s,a) - \lambda_k\, c(s,a)$. The new value for $\lambda_k$ in each iteration is $\lambda_k = \lambda_{k-1} + \frac{\alpha_0}{k}\left(V_C^{k,*} - \hat{c}\right)$, with $V_C^{k,*}$ being the optimal $V_C$ for the optimal policy with a fixed $\lambda_k$ at data gathering iteration $k$. The data gathering loop is terminated when $\hat{c} - \epsilon \leq V_C^{k,*} \leq \hat{c}$. Here, $\alpha_0$ and $\epsilon$ are tunable hyper-parameters.

**Proposition 1.** *This iterative optimization process converges to the optimal $\lambda^*$.*

*Proof.* Previous work (Kocsis & Szepesvári, 2006; Silver et al., 2016) shows that the MCTS policy converges to the optimal policy as the number of simulations increases, meaning that in each iteration $k$ we are (asymptotically) guaranteed to find the optimal solution in the $k$-th MDP. Based on this, and on the fact that $\lambda$ is updated following the gradient direction of $V_C^{k,*} - \hat{c}$, convergence to the optimal $\lambda^*$ is achieved (Lee et al., 2018; Tessler et al., 2018; Mankowitz et al., 2020).

As MCTS with upper confidence bounds converges asymptotically to the optimal policy (Kocsis & Szepesvári, 2006), usually a time- or computational budget-limit is used to terminate learning (Silver et al., 2016; Schrittwieser et al., 2020a). As we are interested in a *feasible* solution, we terminate the training process (search for $\lambda^*$ effectively) in the data gathering phase only when enough data has been gathered and the cost constraints are satisfied (see "data gathering" phase in Fig. 1).

### 3.1.1 TRAINING DATA COLLECTION

At this point, we could utilize the available trajectory data collected from applying the optimal policy on the MDP defined by each $\lambda_k$ to train a safety critic and then deploy it in the environment. For the training, we use State–action–reward–state–action (SARSA)(0) algorithm (Sutton & Barto, 2018) (a Temporal Difference (TD) Learning-like method (Sutton, 1988)). Of course, there is always the risk that the resulting critic (thus also the MCTS policy that utilizes it) does not generalize well far from the collected training data (Ross et al., 2011). Ideally, the training data covers the entire state-action space, but with a higher focus on states where selecting a specific action (over others) has a high effect on expected future performance (Rexakis & Lagoudakis, 2012; Kumar et al., 2022) or cost violations/feasibility in our case.

**Definition 3.** *A state $s$ is said to be cost-non-critical if*

$$\forall a \in A, \quad \min_{a'} Q_c^\pi(s, a') \leq Q_c^\pi(s, a) \leq \hat{c} \quad or \quad \hat{c} \leq \min_{a'} Q_c^\pi(s, a') \leq Q_c^\pi(s, a) \tag{5}$$

In other words, in cost-non-critical states, selecting any action under the applied policy $\pi$ does not lead (in expectation) to a change in the constraint/threshold violation (positive or negative).[2] Even though having more training data from cost-critical states is desirable, these do not frequently occur in trajectories generated by any policy $\pi$ (see also the discussion in Kumar et al. (2022)).

In our case though, as the value of $\lambda$ is iteratively adapted in the "data gathering" phase shown in Fig. 1, state-action pairs around the constraint-switching hypersurface are collected. The use of all available data (generated by different policies $\pi_k$ as a result of all values of $\lambda_k$) for the safety critic training ("model training" phase in Fig. 1), thus ensures that a large collection of state-action pairs from both critical and non-critical states are available.[3] This safety critic is in turn re-used in the MCTS planner of a new $\lambda$-tuning cycle, until a robust safety critic leading to a *feasible solution* is produced (as evaluated in the last phase shown in Fig. 1).

**Assumption 1.** *SARSA(0) estimates with sufficient accuracy the true state-action value function for any given policy $\pi$.*

There have been various formal results on the convergence properties of SARSA-like algorithms both for tabular (Singh et al., 2000) and linear function (Zhang et al., 2023) representations, as well as successful implementations using neural networks as approximators (Elfwing et al., 2018). It is therefore safe to assume that given representative training samples, Assumption 1 holds.

**Proposition 2.** *Let $B = \{(s, a)|s \in S \text{ and } a \in A\}$ be the set of all state-action pairs for a given MDP. Then, there exist $B_p \subseteq B$, a set of state-action pairs for which the trained safety critic would over-estimate the expected discounted cumulative cost and $B_n \subseteq B$, a set of state-action pairs for which the trained safety critic would under-estimate it.*

What Proposition 2 indicates is that the trained safety critic will under-estimate or over-estimate the expected cost of *every* state-action pair defined in the underlying MDP of the *high-fidelity* simulator. If this was not true, this would imply that the safety critic provides the *perfect* prediction at least

---

[2]Note that a similar discussion, under the concept of $\epsilon$-reducible datasets (or parts of datasets), also exists in safe/constrained offline reinforcement learning approaches (Liu et al., 2023).

[3]With this data mixture we train the safety critic using $(s, a)$ samples that have different cost-targets (due to different $\lambda$'s), some of them over- or under-estimating the "true" cost. We could e.g. give higher weight to data from trajectories where the value of $\lambda$ was close to $\lambda^*$, but we observed that using an ensemble of safety critics (see Sec. 3.1.2) combined with using the latest safety critic in each "data gathering" outer loop, leads to "correct" cost data being predominant and thus to a robust final safety critic, possibly at the cost of collecting more data.

for *all* state-action pairs. This is rarely the case, both due to numerical precision issues, as well as due to the utilization of the low-fidelity simulator in the MCTS planner, which potentially predicts sequences of safe or unsafe next states that are different compared to the actual ones, especially for state-action pairs that are far from the terminal states.

**Corollary 1.** *The overall training process of the safety critic, illustrated in Fig. 1, converges to a feasible solution of the constrained optimization problem defined in (1).*

*Proof sketch.* As discussed before, the inner training loop will always converge to the optimal solution for the $k-$th MDP. In case the safety critic over-estimates the expected cost, it prunes the corresponding branch in the MCTS tree. This leads to a *safe*, but potentially *conservative (i.e., non-optimal)* behavior. In case of under-estimation, the respective branch can be traversed and a non-safe trajectory is performed at the high-fidelity simulator. Since these data are used in subsequent safety critic training iterations, the new versions of the safety critic will no longer under-estimate the cost, progressively for all the $(s, a) \in B_n$ pairs (as defined in Proposition 2) that are visited in the high-fidelity simulator and there will be no constraint violations, i.e., we will have a *feasible solution*.

### 3.1.2 ROBUSTNESS TO MODEL MISMATCH

Even if the safety critic has been evaluated as safe during training, there is a chance that subpar performance is observed at deployment as environment dynamics are likely different between the training (source) and the deployment (target) domain. Since MCTS explores the state space exhaustively during the online planning phase, some state-action pairs encountered during planning are likely to be *out-of-distribution*, i.e., differ compared to the trajectories encountered during training.

More formally, we have two main sources of inaccurate safety critic predictions: the *aleatoric* and the *epistemic* uncertainty. The former is inherent in the training data (e.g., due to the stochastic nature of the transition model) and the latter is due to the lack of training data (e.g., it could appear as extrapolation error) – see for example (Chua et al., 2018) for a more formal discussion.

To mitigate the effect of both uncertainty sources, instead of training a single safety critic, we train an ensemble. The individual members of the ensemble have the form of neural networks and approximate the state-action cost function. We denote this ensemble safety critic as $\hat{Q}_{sc}^*(s, a)$. The trainable parameters of each member of the ensemble are optimized to minimize the mean-squared Bellman error which uses a low variance one-step TD-target. The aggregated ensemble output $(\hat{\mu}, \hat{\sigma})$ provides a mean and a standard deviation computed from the individual member's outputs, which we then use within MCTS. Hence, the safety critic output with an ensemble standard deviation greater than a set threshold $\hat{\sigma} > \sigma_{max}$ can be used to identify and ignore those samples and predictions.

---

**Algorithm 1: C-MCTS** | Using a learned safety critic in MCTS.

1   $\mathcal{N}_{root}$ : Root node representing the current state, $s_0$.
2   $\mathcal{N}_{leaf}$ : Selected leaf node with state $s_t$.
3   $\mathcal{P}$ : Traversed path from the root node to the leaf node $(s_0, a_0, s_1, a_1, ..., a_{t-1}, s_t)$.
4   **repeat**

       `// SELECTION`
5      $\mathcal{P}, \mathcal{N}_{leaf} \leftarrow \text{SELECT}(\mathcal{N}_{root})$ `// selection using UCT algorithm.`
       `// EXPANSION`
6      i. Get safety critic outputs $(\hat{\mu}, \hat{\sigma})$ for all actions $a_t \in A$ from $\mathcal{N}_{leaf}$.
7      ii. Identify feasible actions i.e. $A_{\text{feasible}} = \{a_t : \hat{\sigma}_{a_t} \leq \sigma_{max}\}$.
8      iii. Calculate the cost estimate $\hat{Q}_{sc}^*(s_t, a_t)$ for actions $a_t \in A_{\text{feasible}}$.
9      iv. Define: $C_{path} = c(s_0, a_0) + \gamma \cdot c(s_1, a_1) + ... + \gamma^{t-1} \cdot c(s_{t-1}, a_{t-1})$
10     v. Identify unsafe actions i.e. $A_{\text{unsafe}} = \{a_t \in A_{\text{feasible}} : C_{path} + \gamma^t \cdot \hat{Q}_{sc}^*(s_t, a_t) > \hat{c}\}$.
11     vi. Expand tree for branches with safe actions, $a_t \in A \setminus A_{\text{unsafe}}$.
       `// SIMULATION`
12     $\hat{V}_R \leftarrow \text{ROLLOUT}(\mathcal{N}_{leaf})$ `// Get Monte Carlo reward estimate.`
       `// BACKPROPAGATION`
13     $\text{BACKUP}(\hat{V}_R, \mathcal{P})$ `// Update tree statistics.`
14   **until** *maximum number of planning iterations is reached*

---

## 3.2 DEPLOYMENT

The trained safety critic is used during the expansion phase in MCTS, see Alg. 1 – the other phases (selection, simulation, backpropagation) are identical to vanilla MCTS. At the expansion phase, we try to expand the search tree from the leaf node along different branches corresponding to different actions. First, based on the safety critic's output we filter out predictions that we cannot trust (corresponding to high ensemble variance) and create a reduced action set (lines 6-7). The safety of each action from this set is evaluated based on the safety critic's output predicting expected cumulative costs from the leaf. This is summed up with the one-step costs stored in the tree from the root node to the leaf node. If this total cost estimate is greater than the cost constraints ($\hat{c}$), then we prune the corresponding branches, while other branches are expanded (lines 8-11). These steps, when repeated over multiple planning iterations create a search tree exploring a safe search space. As exploration is mostly limited to a safe search space, C-MCTS manages to construct a deeper search tree with fewer planning iterations compared to CC-MCP. We have systematically observed in our results that C-MCTS operates safely closer to the cost-constraint and collects higher rewards with fewer planning iterations as compared to CC-MCP.

## 4 EVALUATION

We evaluate our method by comparing its performance with our baseline CC-MCP (Lee et al., 2018) on *Rocksample* and *Safe Gridworld* environments (see Sec. A.1). We also present insights on the planning efficiency of the proposed algorithm, as well as its sensitivity to different design options, such as the length of planning iterations during training and the values of the ensemble threshold during deployment. We also provide insights on its robustness to model mismatch.

A detailed description of the environments can be found in A.1. Previous work using MCTS (or tree-based planners in general), has been able to address problems with extremely large state (e.g., Atari, Go, Chess and Sogi in Schrittwieser et al. (2020b)) and action spaces (e.g., see representative references in Afsar et al. (2022) for approaches that facilitate some form of clustering of potential actions). Considering that scalability of MCTS has already been addressed, we have tried – following relevant recent discussions in the community (Togelius & Yannakakis, 2023) – to define environments that are computationally manageable while still providing insights on properties and the quality of the final, feasible solution of our algorithm, with respect to the constraint formulation. This not only enables the reproducibility of our work, but also strengthens the statistical significance of our results, as we evaluate with a large number of seeds for the trained safety critic (Henderson et al., 2018).

We measure the performance of the agent on different sizes and complexities of *Rocksample* environments (Sec. A.1.1), with C-MCTS, CC-MCP, and vanilla MCTS (for penalized reward function with known $\lambda^*$). C-MCTS obtains higher rewards than CC-MCP (see Fig. 2, top row). The reward for C-MCTS increases with the number of planning iterations. Also, the agent operates consistently below the cost-constraint (see Fig. 2, middle row), close to the safety boundary while CC-MCP acts conservatively w.r.t costs and performs sub-optimally w.r.t. rewards. Also, costs incurred in each episode vary greatly with different environment initializations. This is mitigated with C-MCTS since cost estimates with TD learning have a lower variance than Monte Carlo cost estimates. Hence, the total number of cost violations is lower for C-MCTS compared to the other methods, in spite of operating closest to the safety constraint (see Fig. 2, bottom row). Vanilla MCTS obtains higher rewards than CC-MCP. This is because $\lambda^*$ is known, and unlike CC-MCP, doesn't require tuning online. MCTS operates close to the cost-constraint but has a high number of cost violations. C-MCTS, when compared to vanilla MCTS, is safer and obtains equally high rewards or in some cases even acts better (e.g., Rocksample$(11, 11)$).

**Planning efficiency.** We compare the planning efficiency of the three methods for the same set of experiments addressed previously. The comparison is done based on the depth of the search tree, given a specific computational budget (i.e., a fixed number of planning iterations). This comparison is qualitative and is used to evaluate the effectiveness of different planning algorithms.

Fig. 3 shows that C-MCTS performs a more narrow search for the same number of planning iterations. The peak tree depth when averaged over 100 episodes is the highest for C-MCTS. In C-MCTS the exploration space is restricted using the safety critic, and this helps in efficient planning. In Rocksample$(11, 11)$ the peak tree depth of CC-MCP is high in spite of having a sub-optimal performance. This is probably because the Lagrange multiplier in CC-MCP gets stuck in a local maximum and is unable to find a globally optimal solution.

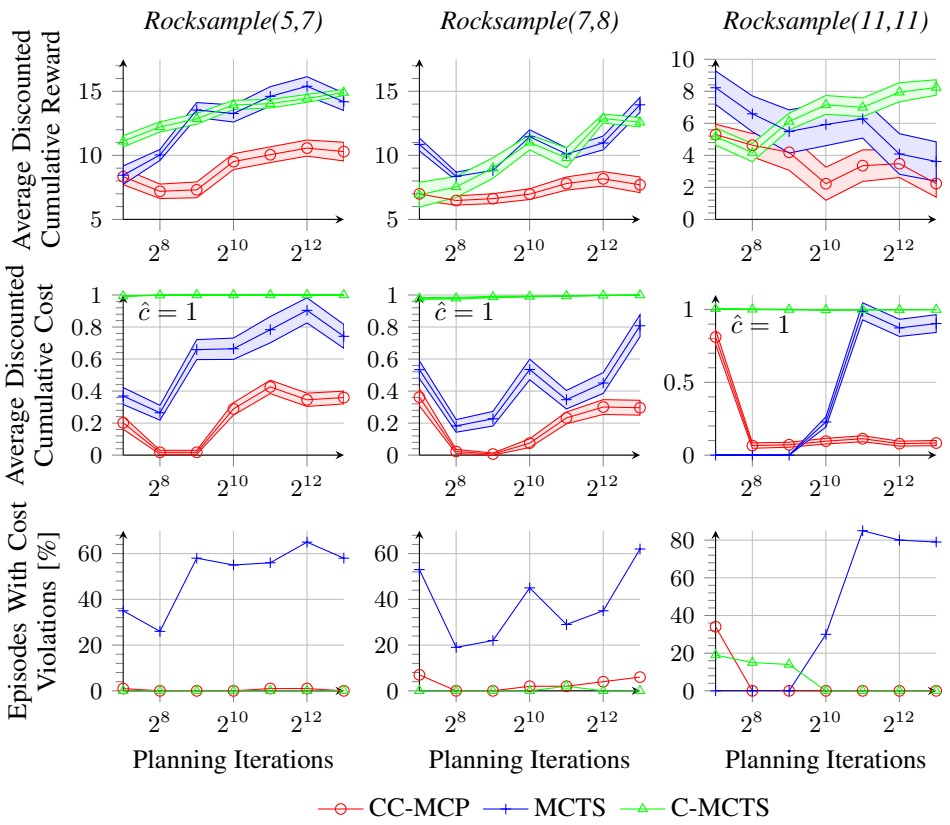

Figure 2: Comparing performance of C-MCTS, MCTS, and CC-MCP on different configurations of *Rocksample* environments evaluated on 100 episodes.

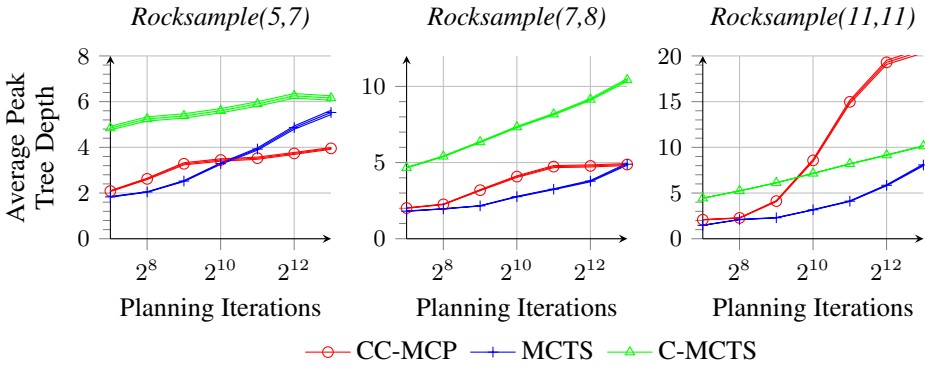

Figure 3: Maximum depth of the search tree for C-MCTS, MCTS and CC-MCP on different rock-sample configurations averaged over 100 episodes.

## 4.1 ROBUSTNESS TO MODEL MISMATCH

**MCTS planner model.** Online planners often resort to approximate models of the real world for fast planning and to account for real-time decision requirements. Those model imperfections can lead to safety violations while a high-fidelity model during deployment is infeasible due to computational constraints. We resolve this dilemma by learning safety constraints before deployment from a simulator that has a higher fidelity compared to the planning model. The benefit of such an approach is shown using a synthetically constructed *Safe Gridworld* scenario (see Sec. A.1.2).

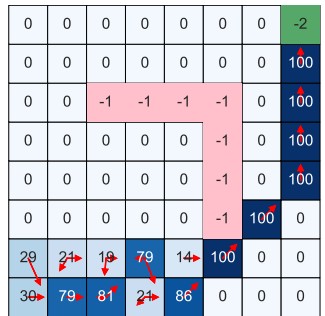
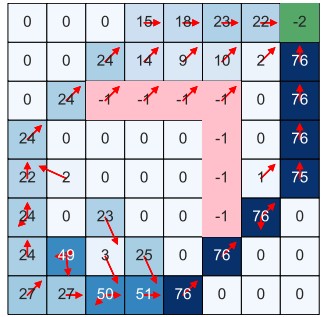

(a) C-MCTS with $0\%$ cost violations.      (b) CC-MCP with $11\%$ cost violations.

Figure 4: State visitations aggregated over 100 episodes. The length of the arrows is proportional to the number of action selections. Values of -1 and -2 denote unsafe cells and the goal cell, respectively.

In this setup, we use a planning simulator that models the dynamics approximately, and a training simulator (for the safety critic) that captures the dynamics more accurately. In the planning simulator, all transition dynamics are accurately modeled, except the blue squares with winds (Fig. 6 right). The transitions here are determined by the action selection (stochasticity due to wind is not considered). The training simulator models the transitions in these regions more accurately, but with some errors. The agent in the blue squares moves down with a probability of $0.25$, as compared to the real-world configuration where the probability is $0.3$. We trained and evaluated C-MCTS for $2^9$ and CC-MCP for $2^{20}$ planning iterations. The latter was set to a higher planning budget to let it converge.

Fig. 4 shows the number of state visitations of C-MCTS (left) and CC-MCP (right). The acCC-MCP agent takes both of the possible paths (going to the top and to the right), avoiding the unsafe region (in pink) to reach the goal state, which is optimal in the absence of the windy squares, but here it leads to cost violations due to inaccurate cost estimates.[4] C-MCTS on the other hand only traverses through the two right-most columns to avoid the unsafe region, as the safety critic being trained using the high-fidelity simulator identifies the path from the top as unsafe, which leads to zero cost violations.

**Accuracy of the training simulator.** We study the performance of C-MCTS when trained on imperfect simulators. On the Rocksample environment, the sensor characteristics measuring the quality of the rock are defined by the constant $d_0$ (see Sec. A.1.1). We overestimate the sensor accuracy in our training simulator by choosing $d_0^{sim}$ with error $\Delta d_0$ and observe the safety of the agent in the real world when trained on simulators with different values of $\Delta d_0$.

Fig. 5 (right column) shows the results. The values of $\Delta d_0$ set to 10 and 40 correspond to a maximum prediction error of $11.7\%$ and $32.5\%$, respectively. When $\Delta d_0 = 40$ the agent operates at a greater distance from the cost-constraint. The reason for cost violations is that the safety critic has been trained to place too much trust in the sensor measurements due to the simulation-to-reality gap. With a smaller gap ($\Delta d_0 = 10$) the agent performs safer.

## 4.2 EFFECT OF HYPER-PARAMETER SELECTION ON SAFE BEHAVIOR

We vary hyper-parameters in the training and deployment of a safety critic and identify key parameters for safety. We conducted our experiments on Rocksample$(7, 8)$ and averaged the results over 100 runs. We observed that C-MCTS is sensitive to the length of the planning horizon during training and the ensemble threshold used during deployment. To study the effect of these parameters we optimized the other algorithmic parameters, i.e., $\alpha_0$ (initial step size to update $\lambda$) and $\epsilon$ (termination criterion for training loop), with a grid search. For each of the experiments we then selected a reasonably performing configuration (i.e., Sec. 4.2: $\alpha_0=4$, $\epsilon=0.1$; Sec. 4.2: $\alpha_0=8$, $\epsilon=0.3$; Sec. 4.1: $\alpha_0=1$, $\epsilon=0.1$) and ablated the respective hyperparameters, i.e., planning horizon in training, $\sigma$, and $d_0$.

**Length of planning horizon during training.** We conduct experiments with different sets of hyper-parameters to study the effect of using different planning iterations during the safety critic training, and evaluate the performance of the agent during deployment. From Fig. 5 (left column)

---

[4]Of course also the variance could play a minor role but we designed the setup to focus on the dynamics mismatch between the planner and the actual environment, which is much more prevalent here.

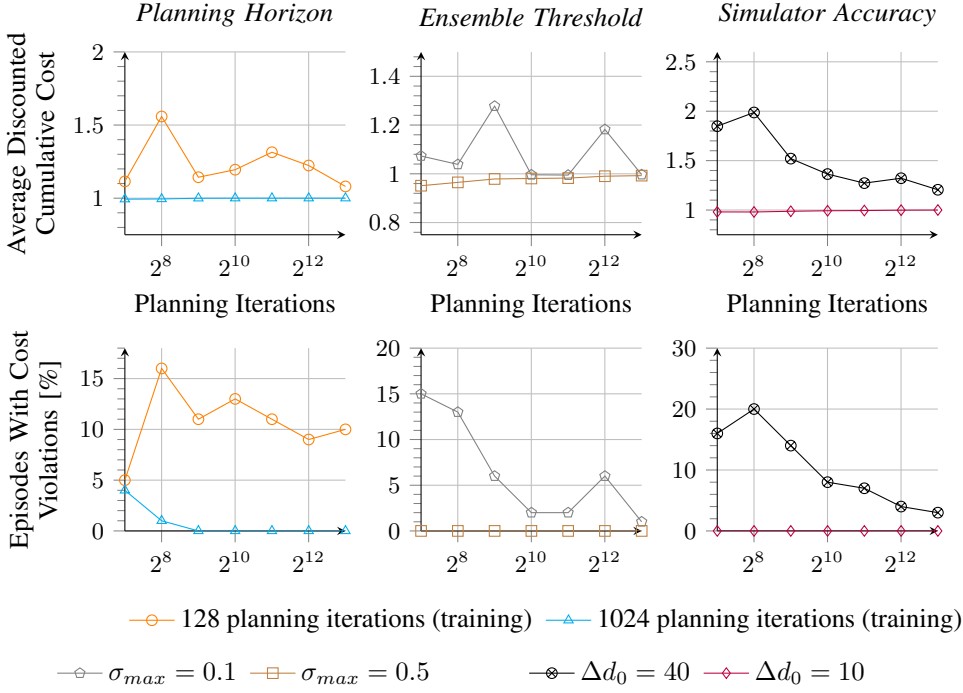

Figure 5: Comparing safety for different training/deployment strategies, i.e., using different planning horizons during training (left), deploying with different ensemble thresholds (middle), and collecting training samples from simulators of different accuracies (right).

we can observe that the safety critic trained with a longer planning horizon operates closer to the safety boundary. This is because the safety critic predicts costs for a near-optimal policy and hence discerns the safety boundary more accurately. The safety critic trained with a smaller planning horizon estimates costs from a sub-optimal policy leading to cost violations during deployment.

**Ensemble threshold during deployment.** We study the effect of using different standard deviation thresholds ($\sigma_{max} = 0.1$ and $\sigma_{max} = 0.5$) in the neural network ensemble during deployment. Fig. 5 (middle column) shows that the cost incurred exceeds the cost-constraint if $\sigma_{max} = 0.1$, but the agent performs safely within the cost-constraint with a far lesser number of cost violations if $\sigma_{max} = 0.5$. We prune unsafe branches during planning only when the predictions between the individual members of the ensemble align with each other. Setting $\sigma_{max} = 0.1$ is a tight bound resulting in most of the predictions of the safety critic being ignored. Using a higher threshold with $\sigma_{max} = 0.5$ ensures that only large mismatches between the predictions of the individual members (corresponding to out-of-distribution inputs) are ignored, and the rest are used during planning. This results in the agent performing safely within the cost-constraint, but not too conservatively.

## 5 CONCLUSION

We presented an MCTS-based approach to solving CMDPs that learns the cost-estimates in a pre-training phase from simulated data and that prunes unsafe branches of the search tree during deployment. In contrast to previous work, C-MCTS does not need to tune a Lagrange multiplier online, which leads to better planning efficiency and higher rewards. In our experiments on Rocksample environments, C-MCTS achieved maximum rewards surpassing previous work for small, medium, and large-sized grids with increasing complexity, while maintaining safer performance. As cost is estimated from a lower variance TD target the agent can operate close to the safety boundary with minimal constraint violations. C-MCTS is also suited for problems using approximate planning models for fast inference, but having to adhere to stringent safety norms. Our Safe Gridworld setup demonstrates that even with an approximate planning model, the notion of safety can be learned separately using a more realistic simulator, resulting in zero constraint violations and improved safety.

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

## A SUPPLEMENTARY MATERIAL

### A.1 ENVIRONMENTS

#### A.1.1 ROCKSAMPLE

The environment is defined as a grid with $n \times n$ squares with $m$ rocks randomly placed, some being good and others bad (see Fig. 6, left). A specific Rocksample setup is defined by the nomenclature Rocksample$(n, m)$. A rover (agent) starting from the left is tasked to collect as many good rocks as possible and exit the grid to the right. The positions of the rocks are known in advance, but the quality of the rocks is unknown. The agent can move up, down, right, and left, sample a rock, or make measurements to sense the quality of a rock. The total number of possible actions is hence $5 + m$. The agent is equipped with a noisy sensor to measure the quality of a rock with a probability of accuracy $(2^{-d/d_0} + 1)/2$, where $d$ is the Euclidean distance of the agent from the corresponding rock and $d_0$ is a constant. The number of measurements that the agent can perform is constrained. Trying to maximize rewards (collecting good rocks) with constraints (number of sensor measurements) encourages the agent to use a limited number of measurements at a reasonable proximity to the rocks, wherein the sensor readings can be trusted. At each time step the agent observes its own position and the positions of the rocks with the updated probabilities.

We formulate the task within the CMDP framework by additionally defining a reward structure, cost function, and a cost-constraint. The agent is rewarded a +10 reward for exiting the grid from the right or for collecting a good rock. A -10 penalty is received for each bad rock collected, and a -100 penalty is given when the agent exits the grid to the other sides or if the agent tries to sample a rock from an empty grid location. The agent incurs a +1 cost when measuring the quality of a single rock. The discounted cost over an episode cannot exceed 1, and this is the cost-constraint. The discount factor $\gamma$ is set to 0.95.

#### A.1.2 SAFE GRIDWORLD

We additionally propose a new problem: *Safe Gridworld*. The environment is defined as $8 \times 8$ grid where an agent from the bottom left region is tasked to find the shortest path to reach the top right square avoiding unsafe squares on the way (see Fig. 6, right). The agent can move to the neighboring squares and has a total of 9 action choices. The transition dynamics in all squares are deterministic except the 8 squares at the top which are stochastic. These squares have winds blowing from the top to the bottom forcefully pushing the agent down by one square with a probability of 0.3, independent of the action chosen by the agent (we vary this probability to account for simulator mismatch in the experiment in Sec. 4.1). Otherwise, the transition is guided by the agent's action.

The agent receives a reward of +100 on reaching the goal state, a -1000 penalty for exiting the grid, and a -1 penalty otherwise until the terminal state is reached. Entering an unsafe square incurs a cost of +1. The agent should only traverse safe squares, and the discounted cost over an episode is 0. The cost-constraint imposes this as a constraint and is set to 0. The discount factor $\gamma$ is set to 0.95.

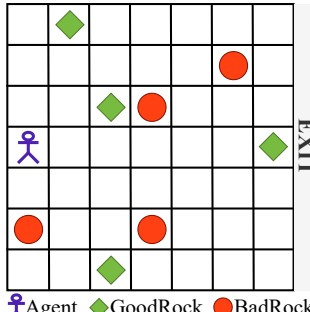 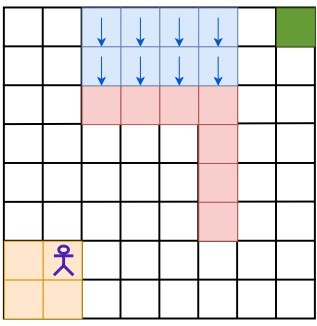

Figure 6: Environments: (left) exemplary Rocksample$(7, 8)$ environment, i.e., a $7 \times 7$ rocksample environment with 8 rocks randomly placed; (right) exemplary Safe Gridworld environment, where the colors denote start cells (yellow), the goal cell (green), unsafe cells (pink), and windy cells (blue).

## A.2  TRAINING DETAILS & COMPUTE

The training and evaluation were conducted on a single Intel Xeon E3-1240 v6 CPU. The CPU specifications are listed below.

| Component | Specification |
|---|---|
| Generation | Kaby Lake |
| Number of Cores | 4 |
| Hyper-Threading (HT) | Disabled |
| Base Frequency | 3.70 GHz |
| RAM | 32 GB |
| SSD | 960 GB |

Table 1: Specifications of Intel Xeon E3-1240 v6

No GPU accelerators were used as the C-MCTS implementation was not optimized for efficient GPU resource utilization. The hyperparameters chosen for training the safety critic in the primary results (Fig. 2) are summarized in Table 2.

| Environment | $\alpha_0$ | $\epsilon$ | $\sigma_{max}$ | Planning Iterations |
|---|---|---|---|---|
| Rocksample$(5, 7)$ | 8 | 0.1 | 0.5 | 1024 |
| Rocksample$(7, 8)$ | 4 | 0.1 | 0.5 | 1024 |
| Rocksample$(11, 11)$ | 12 | 0.1 | 0.5 | 512 |
| Safe Gridworld | 10 | 0.1 | 0.2 | 512 |

Table 2: Key hyperparameters to train the safety critic.

## A.3  PLANNING COST TO ACHIEVE HIGH REWARDS: C-MCTS VS CC-MCP

| Environment | Method | Performance | |
|---|---|---|---|
| | | Number of planning iterations | Discounted Reward |
| $N/A$: Performance not achieved | | (*): Additional evaluation environment | |
| Rocksample(5,7) | CC-MCP | $2^{20}$ | 13.72 |
| | C-MCTS | $2^{10}$ | 13.93 |
| Rocksample(7,8) | CC-MCP | $2^{20}$ | 9.83 |
| | C-MCTS | $2^{10}$ | 11.0 |
| Rocksample(11,11) | CC-MCP | $2^{20}$ | 5.26 |
| | C-MCTS | $2^{10}$ | 7.14 |
| Rocksample(15,15)* | CC-MCP | $N/A$ | $N/A$ |
| | C-MCTS | $2^{8}$ | 14.29 |

Table 1: Comparing planning iterations of C-MCTS and CC-MCP at equivalent reward levels.

## A.4  COMPUTATIONAL COST COMPARISON

In terms of computational cost per simulation across the different algorithmic phases:

- Selection: CC-MCP is the most computationally expensive, requiring more operations to select the best child node. MCTS and C-MCTS have identical operation counts.

- Expansion: C-MCTS incurs additional costs due to the safety critic's prediction. MCTS and CC-MCP require no additional computation during expansion.

- Backpropagation: CC-MCP backs up Q-values for both reward and cost, while MCTS and C-MCTS back up only Q-values for reward.

- Rollout: Computational cost is identical for C-MCTS, MCTS, and CC-MCP.

C-MCTS and MCTS algorithms were implemented in Python, while the benchmark CC-MCP uses a C++ implementation. Comparing actual execution times was unfair since our implementation was not optimized for hardware efficiency. Also, such an optimization would highly depend on the hardware platform (e.g. CPUs vs GPUs). For instance, added cost in C-MCTS's expansion phase is highly parallelizable, with good potential for effective GPU utilization. So, for our analysis we instead

compare the number of simulations (planning iterations) required by each algorithm as a performance metric.

## A.5 BROADER IMPACT

While C-MCTS mitigates the reliance on the planning model to meet cost constraints through pre-training in a high-fidelity simulator, there may still be sim-to-reality gaps when learning cost estimates. This introduces the possibility of encountering unforeseen consequences in real-world scenarios. In the context of using C-MCTS in a human-AI interaction task, if minority groups are not adequately represented in the training simulator, inaccurate cost estimates might lead to potential harm to humans. However, C-MCTS addresses these gaps more effectively than previous methods by leveraging a more relaxed computational budget during the training phase (fast inference is only required during deployment). This allows more accurate modeling of the real world to include rare edge scenarios.

