# OpenReview forum: "C-MCTS: Safe Planning with Monte Carlo Tree Search"
_ICLR.cc/2024/Conference — Submitted to ICLR 2024_

### Official Review · Reviewer_q4t7 · 2023-10-27

**Soundness:** 2 fair
**Presentation:** 3 good
**Contribution:** 2 fair
**Rating:** 5
**Confidence:** 3

**Summary:**

This paper introduces Constrained MCTS (C-MCTS), an MCTS-based novel approach for solving CMDP decision-making tasks. This approach avoids constraint violations by pre-training a safety critic before agent deployment with TD learning using simulator data. The pre-trained safety critic eliminates the need for tuning a Lagrange multiplier online. The experiments in two environments, Rocksample and Safe Gridworld, show that C-MCTS outperforms existing approaches. It prunes unsafe branches of the search tree during deployment, achieving higher rewards while ensuring safety. Most importantly, it is less susceptible under the model mismatch between the simulator and the real-world environment.

**Strengths:**

This paper proposes a novel and straightforward method to pre-train a safety critic using the TD-based method. By negating the need to tune the Lagrange multiplier online, the pre-training phase allows the use of a more computationally expensive high-fidelity simulator, thereby producing better results. Besides, it also allows the ensemble method to be applied to have more robust results.

**Weaknesses:**

While the method is novel and straightforward, there are two major concerns as follows.

First, the time complexity of C-MCTS seems higher than normal MCTS. In the expansion of C-MCTS, the safety critic is used to evaluate all actions. This greatly increases the computing resources when applying in the large action space domains or using more MCTS simulations. In addition, considering C-MCTS uses additional resources to tune a Lagrange multiplier, the comparison between C-MCTS and CC-MCP is unfair. The authors should provide the detailed time complexity for C-MCTS, and the time for the pre-training phase for comparison.

Second, the experimental environments are quite simple, so it is hard to convincingly evaluate the out-of-distribution scenarios where states are completely unseen during deployment. The author should add experiments on more complex environments, e.g., the MuJoCo physics simulator.

**Questions:**

* P2, Figure 1. As introduced in Section 3.1, the training phase uses MCTS to solve MDP, which should also be added to the figure for better understanding.
* P2, Section 2.1. Use bold font to emphasize sets (especially for $C$ and $\hat{c}$) in definitions and equations for clear reading. All policies $\Pi$ is undefined in equation (1).
* P3, Section 2.2, "Lee et al. (2018) proposed Cost-Constrained Monte Carlo Planning (CC-MCP)". This is incorrect since Lee et al. proposed Cost-Constrained POMCP (CC-POMCP) instead of CC-MCP. Although it applies to CMDPs, the authors should revise the statement to clarify this.
* P3, Section 3.1. Clarify what "e.g., in simulation" means. Does it mean to vary parameters during MCTS simulation in the pre-training phase? In addition, $V_{C}^{k,*}$ is undefined.
* P4, Section 3.1.1, Definition 3. In equation (5), it seems meaningless to define $\mathop{\min}_{a'} {Q^{\pi}_{c}(s,a')} <= Q^{\pi}_{c}(s,a)$ when $\forall a' \in A$.
* P4, Proposition 2, "This is not only numerically infeasible, but it is also due to the utilization of the low-fidelity simulator in the MCTS planner." It is still unclear why this is numerically infeasible. In common sense, even if a low-fidelity simulator is employed, it is still likely that the critic can provide perfect estimation for some state-action pairs, especially for those near the terminal states.
* P5, Section 3.1.1, Corollary 1, "As discussed before, the inner training loop will always converge to the optimal solution for the k−th MDP." The convergence happens when there are sufficient training simulations, as stated in Proposition 1. However, did the conducted experiment have sufficient simulations to achieve the convergence, i.e., the optimal solution?
* P6, Section 4, "the selected environments clearly "emulate" the complexity that stems from domains with large state/action spaces (Schrittwieser et al., 2020b; Afsar et al., 2022)." Please clarify the reason for citing these two papers here. They do not seem to have any discussion about the selected environments.
* P8, Section 4.2, "with a grid search". Did the optimized $\alpha_0$ and $\epsilon$ highly affected the final performance? The authors should provide the grid search results in the appendix for reference.

---

> ### Author Response · Authors · 2023-11-16
>
> First and foremost, we would like to thank you for your valuable feedback. It helped us a lot to clarify points in our paper!
>
> > First, the time complexity of C-MCTS seems higher than normal MCTS. In the expansion of C-MCTS, the safety critic is used to evaluate all actions. This greatly increases the computing resources when applying in the large action space domains or using more MCTS simulations. In addition, considering C-MCTS uses additional resources to tune a Lagrange multiplier, the comparison between C-MCTS and CC-MCP is unfair. The authors should provide the detailed time complexity for C-MCTS, and the time for the pre-training phase for comparison.
>
> Thank you for your question! You are right that we must evaluate the safety critic for each action the agent might take in a state. However, as our safety critic is a neural network that output cost estimates for all the available actions through single forward pass (just like in a DQN) its constant time and only little overhead in addition to CC-MCP and MCTS.
>
> Please also find below a more rigorous explanation of incurred costs that we also lined out to reviewer q5zX and that we have added to the appendix of the paper:
>
> In terms of computational cost per simulation across the different algorithmic phases:
> - Selection: CC-MCP is the most computationally expensive, requiring more operations to select the best child node. MCTS and C-MCTS have identical operation counts.
> - Expansion: C-MCTS incurs additional costs due to the safety critic's prediction. MCTS and CC-MCP require no additional computation during expansion.
> - Backpropagation: CC-MCP backs up Q-values for both reward and cost, while MCTS and C-MCTS back up only Q-values for reward.
> - Rollout: Computational cost is identical for C-MCTS, MCTS, and CC-MCP.
> C-MCTS and MCTS algorithms were implemented in Python, while the benchmark CC-MCP uses a C++ implementation. Comparing actual execution times was unfair since our implementation was not optimized for hardware efficiency. Also, such an optimization would highly depend on the hardware platform (e.g. CPUs vs GPUs). For instance, added cost in C-MCTS's expansion phase is highly parallelizable, with good potential for effective GPU utilization. So, for our analysis we instead compare the number of simulations (planning iterations) required by each algorithm as a performance metric.
>
> > Second, the experimental environments are quite simple, so it is hard to convincingly evaluate the out-of-distribution scenarios where states are completely unseen during deployment. The author should add experiments on more complex environments, e.g., the MuJoCo physics simulator.
>
> We avoided choosing complex environments, since using complex models within the search tree during planning is expensive. Eventhough our benchmark Rocksample has simple dynamics, it has a high dimensional state and large action space. We then progressively increased the domain size (5×5→7×7→11×11) and size of the action space (12→13→16). This increases the search space while being easy in the environment setup and achieving meaningful results.
>
> During pre-training the safety critic is trained ONLY on states observed in the high-fidelity simulator during the episode. It does NOT see states corresponding to nodes of the search tree (in simulation). The states visited in the high-fidelity simulator are a subset of the states visited by the search tree in simulation. Consequently, it is reasonable to anticipate that, during deployment, there is a considerable likelihood of the safety critic encountering states within the search tree that were not observed during training.
>
> > P2, Figure 1. As introduced in Section 3.1, the training phase uses MCTS to solve MDP, which should also be added to the figure for better understanding.
>
> Thank you for this suggestion. We changed the Figure accordingly.
>
> > P2, Section 2.1. Use bold font to emphasize sets (especially for  C and \hat{c}) in definitions and equations for clear reading. All policies $\Pi$ is undefined in equation (1).
>
> Thank you for pointing this out. Following the notation convention of [1], $\mathbf{C} = \\{C_m\\}\_{1\ldots M}$ is a set of $M$ non-negative cost functions, with $\mathbf{\hat{c}} = \{c_m\}_{1\ldots M} \in [0,1]$ their respective thresholds. In the text, we assume only one constraint function $C$ with its respective threshold $\hat{c}$, to simplify the notation.
>
> In any case, we have re-written this point more clearly in the new version of the text.
>
> [1] Lee, Jongmin, et al. "Monte-Carlo tree search for constrained POMDPs." Advances in Neural Information Processing Systems 31 (2018).

---

> ### Author Response · Authors · 2023-11-16
>
> > P3, Section 2.2, "Lee et al. (2018) proposed Cost-Constrained Monte Carlo Planning (CC-MCP)". This is incorrect since Lee et al. proposed Cost-Constrained POMCP (CC-POMCP) instead of CC-MCP. Although it applies to CMDPs, the authors should revise the statement to clarify this.
>
> Thank you for pointing this out. We have corrected this in the new version of the paper.
>
> > P3, Section 3.1. Clarify what "e.g., in simulation" means. Does it mean to vary parameters during MCTS simulation in the pre-training phase? In addition,  V_{C}^{k, *} is undefined.
>
> Good point! The correct sentence is: “Instead of tuning the Lagrange multiplier online, C-MCTS varies this parameter in a pre-training phase (in simulation) to obtain different sets of trajectories with different safety levels (“data gathering” in Fig. 1).
>
> Our approach is as follows: we have a low- and a high-fidelity simulator that approximate a real-world problem/environment. The pre-training phase (“data gathering” phase in Figure 1) consists of varying the Lagrange multiplier \lambda and finding the optimal (MCTS) policy and V_C for this value of \lambda (“solve” the MDP). Here, the low-fidelity simulator is used for the planning phase of MCTS, while the high-fidelity simulator is used for training the cost approximation V_C.
>
> Also, we have added the definition for $V_{C}^{k, *}$
>
> > P4, Section 3.1.1, Definition 3. In equation (5), it seems meaningless to define $\mathop{\min}{a'} {Q^{\pi}{c}(s,a')} <= Q^{\pi}_{c}(s,a)\forall a' \in A$.
>
> Correct! As we are striving to find a feasible solution that does not violate the constraints, this first part of Eq. 5 has no effect on the design of a policy leading to minimum (or no) constraint violations. We simply added the term there to highlight clearer the connection to related work [1,2].
>
> [1] Kumar, Aviral, et al. "When should we prefer offline reinforcement learning over behavioral cloning?." arXiv preprint arXiv:2204.05618 (2022).
>
> [2] Liu, Zuxin, et al. "Constrained decision transformer for offline safe reinforcement learning." arXiv preprint arXiv:2302.07351 (2023).
>
> > P4, Proposition 2, "This is not only numerically infeasible, but it is also due to the utilization of the low-fidelity simulator in the MCTS planner." It is still unclear why this is numerically infeasible. In common sense, even if a low-fidelity simulator is employed, it is still likely that the critic can provide perfect estimation for some state-action pairs, especially for those near the terminal states.
>
> Good point. We were thinking along the lines of numerical precision of the estimator, but we agree with the argument that an estimation can be precise for selected classes of problems, especially in state-action pairs near the terminal states. Therefore, we have relaxed the proposition as follows:
>
> “Proposition 2. Let $B = \{(s,a) | s \in S\ \text{and}\ a \in A\}$ be the set of all state-action pairs for a given~\ac{MDP}. Then, there exist $B_p \subseteq B$, a set of state-action pairs for which the trained safety critic would over-estimate the expected discounted cumulative cost and $B_n \subseteq B$, a set of state-action pairs for which the trained safety critic would under-estimate it.
>
> What Proposition 2 indicates is that the trained safety critic will under-estimate or over-estimate the expected cost of every state-action pair defined in the underlying MDP of the high-fidelity simulator. If this was not true, this would imply that the safety critic provides the perfect prediction at least for all state-action pairs. This is rarely the case, both due to numerical precision issues, as well as due to the utilization of the low-fidelity simulator in the MCTS planner, which potentially predicts sequences of safe or unsafe next states that are different compared to the actual ones, especially for state-action pairs that are far from the terminal states.”
>
> Still, we would argue that this does not change the behavior of the algorithm in case of cost over- or under-estimation by the safety critic, hence does not affect Corollary 1.

---

> ### Author Response · Authors · 2023-11-16
>
> > P5, Section 3.1.1, Corollary 1, "As discussed before, the inner training loop will always converge to the optimal solution for the k-th MDP." The convergence happens when there are sufficient training simulations, as stated in Proposition 1. However, did the conducted experiment have sufficient simulations to achieve the convergence, i.e., the optimal solution?
>
> Fair point. As there was a similar point from other reviewers in Proposition 1, we selected to address this there.
> MCTS with upper confidence bounds converges asymptotically to the optimal policy [1], usually a time- or computational budget-limit is used to terminate learning [2,3]. In our case, since we are interested in a feasible solution, we terminate the training process (search for $\lambda^*$ effectively) in the data gathering phase only when enough data has been gathered and the cost constraints are satisfied (see ``data gathering'' phase in Fig. 1).
>
> To solve each of the MDPs we use a sufficiently high number of simulations that ensures a nearly optimal policy (Refer Figure 5, comparing performance for low and high number of simulations). When the costs observed are close to the constraint (but not violated), determined by $\hat{c}-\epsilon \leq V_{C}^{k, *} \leq \hat{c}$,  the iterative process is terminated.
>
> We have adapted the text accordingly to clarify this.
>
> [1] Kocsis, Levente, and Csaba Szepesvári. "Bandit based monte-carlo planning." European conference on machine learning. Berlin, Heidelberg: Springer Berlin Heidelberg, 2006.
>
> [2] Silver, David, et al. "Mastering the game of Go with deep neural networks and tree search." nature 529.7587 (2016): 484-489.
>
> [3]  Schrittwieser, Julian, et al. "Mastering atari, go, chess and shogi by planning with a learned model." Nature 588.7839 (2020): 604-609.
>
> > P6, Section 4, "the selected environments clearly "emulate" the complexity that stems from domains with large state/action spaces (Schrittwieser et al., 2020b; Afsar et al., 2022)." Please clarify the reason for citing these two papers here. They do not seem to have any discussion about the selected environments.
>
> We apologize for not being clear here. Our intention was to conclude the discussion of this paragraph, related to selecting representative problems/environments that are not computationally intensive [1], to be able to run several MCTS planning iterations in a “reasonable” amount of time in moderate hardware infrastructure. Considering the fact that we are evaluating with 100 different seeds for the results to have statistical significance [2], this requirement becomes even more pressing.
>
> Having this in mind, previous work using MCTS (or tree-based planners in general), has been able to address problems with extremely large state (e.g., Atari, go, chess and sogi in [3]) and action spaces (e.g., see representative references in [4] for approaches that facilitate some form of clustering of potential actions). Therefore, we have tried – to the best of our ability – to define “simple” environments that provide insights on properties and the quality of the final, feasible solution of our algorithm, with respect to the constraint formulation, considering that scalability issues of MCTS have already been addressed in previous work.
>
> We have changed the text in the newest version of the paper to better clarify our views.
>
> [1] Togelius, Julian, and Georgios N. Yannakakis. "Choose your weapon: Survival strategies for depressed AI academics." arXiv preprint arXiv:2304.06035 (2023).
>
> [2] Henderson, Peter, et al. "Deep reinforcement learning that matters." Proceedings of the AAAI conference on artificial intelligence. Vol. 32. No. 1. 2018.
>
> [3] Schrittwieser, Julian, et al. "Mastering atari, go, chess and shogi by planning with a learned model." Nature 588.7839 (2020): 604-609.
>
> [4] Afsar, M. Mehdi, Trafford Crump, and Behrouz Far. "Reinforcement learning based recommender systems: A survey." ACM Computing Surveys 55.7 (2022): 1-38.
>
>
> > P8, Section 4.2, "with a grid search". Did the optimized  and  highly affected the final performance? The authors should provide the grid search results in the appendix for reference.
>
> We agree that the results of the grid search should be mentioned in the paper. Unfortunately, we did not keep the full results which is why we just started the grid search again. While the results will not be ready by the rebuttal deadline, we will include them in the final version of the paper.

---

> > ### Comment · Reviewer_q4t7 · 2023-11-23
> >
> > Thank you for addressing my concerns. However, I still think the experiments are not enough to convince readers. Overall, I tend to keep my score.

---

### Official Review · Reviewer_DU49 · 2023-10-31

**Soundness:** 2 fair
**Presentation:** 3 good
**Contribution:** 2 fair
**Rating:** 3
**Confidence:** 5

**Summary:**

The paper proposes an MCTS variant for safety-critical scenarios. The variant expands in leaf nodes only actions that are deemed safe and feasible by a safety-critic model. The model is optimized in a pre-training phase.

The MCTS variant is compared favorably with two other MCTS variants in two grid-worlds.

**Strengths:**

Dealing with safety constraint is probably one of the weaknesses of the RL approaches, and one of the main obstacles for applying RL and planning algorithms like MCTS in real world scenarios. While in many of those scenarios, the algorithms are faced with continuous state/action spaces, tackling the issue in discrete spaces is also important.

The extension of the MCTS for constrained MDP seems fairly reasonable.

**Weaknesses:**

While there is some theoretical work included, these do not offer sufficient guarantees for practical applicability. The proposed algorithm could be a step towards an practical application, but it is not there as it is.

Given that this is a largely empirical article, the experimental evaluation is rather small. The benchmarks are small and fairly simple, while that set of baselines is also limited.

**Questions:**

I understand that there are not many MCTS implementation supporting constrained MDP, but there are a sufficient number of alternative planning algorithms that could have been used as additional baselines.

The size (especially the small number of possible actions) and the limited number of benchmarks are difficult to understand. If this would be a theoretical paper (or at least one offering significantly new algorithmic ideas) using these tasks for illustration would be fine. But, for a mainly empirical paper, the limitation of the experiments are difficult to understand.

---

> ### Author Response · Authors · 2023-11-16
>
> First and foremost we want to thank the review for the time and effort spent our paper. We are sure that the comments help us to improve our paper.
>
> > While there is some theoretical work included, these do not offer sufficient guarantees for practical applicability. The proposed algorithm could be a step towards an practical application, but it is not there as it is.
>
> Thank you for bringing that up. We agree that we did not demonstrate a real-world system using C-MCTS. We share your concern that also with C-MCTS we cannot be sure to have 0 constraint violations during deployment of an agent and we also share your comment that C-MCTS takes a step towards that direction while not being the final solution in any application. As we do not guarantee this, we only claimed to be less susceptible to cost violations than previous work. Please also see our discussion in A.5. We hope through fostering research towards this direction helps to build safe RL systems that can lead to impact in real world applications.
>
> > Given that this is a largely empirical article, the experimental evaluation is rather small. The benchmarks are small and fairly simple, while that set of baselines is also limited.
>
> Thank you for raising this point. Previous work using MCTS (or tree-based planners in general), has been able to address problems with extremely large state (e.g., Atari, go, chess and sogi in [1]) and action spaces (e.g., see representative references in [2] for approaches that facilitate some form of clustering of potential actions). Therefore, we have tried – to the best of our ability – to define “simple” environments that provide insights on properties and the quality of the final, feasible solution of our algorithm, with respect to the constraint formulation, considering that scalability issues of MCTS have already been addressed in previous work.
>
> [1] Schrittwieser, Julian, et al. "Mastering atari, go, chess and shogi by planning with a learned model." Nature 588.7839 (2020): 604-609.
>
> [2] Afsar, M. Mehdi, Trafford Crump, and Behrouz Far. "Reinforcement learning based recommender systems: A survey." ACM Computing Surveys 55.7 (2022): 1-38.
>
> > I understand that there are not many MCTS implementation supporting constrained MDP, but there are enough alternative planning algorithms that could have been used as additional baselines.
>
> We chose baselines that showed competitive results. We excluded CALP, an offline planner,  which was used by Lee et. al. to benchmark CC-MCP since it showed scalability issues in their evaluation. Multi-Objective (MO) MCTS could provide an interesting point of comparison. But, we chose to exclude this since MOMCTS computes a set of Pareto optimal solutions for Multi-Objective MDPs, in contrast to C-MCTS which calculates just a single optima.
>
> > The size (especially the small number of possible actions) and the limited number of benchmarks are difficult to understand. If this would be a theoretical paper (or at least one offering significantly new algorithmic ideas) using these tasks for illustration would be fine. But, for a mainly empirical paper, the limitation of the experiments are difficult to understand.
>
> Even though our benchmark Rocksample has simple dynamics, it has a large-enough state and action space, which lets us study the effects of C-MCTS in various ways. This makes us confident that with the assumption from the answer before it also works in environment which much larger state- and action spaces that require further tweaks (such as AlphaGo-style techniques). As an illustration, we progressively increased the domain size (5×5→7×7→11×11) and size of the action space (12→13→16). This increases the search space while being easy in the environment setup and achieving meaningful results. We avoided choosing complex scenarios, since using complex models within the search tree during planning is expensive.

---

> > ### Comment · Reviewer_DU49 · 2023-11-22
> >
> > I appreciate the reply by the authors, but I still feel that the empirical evaluation is insufficient.

---

### Official Review · Reviewer_EfYF · 2023-10-31

**Soundness:** 1 poor
**Presentation:** 2 fair
**Contribution:** 2 fair
**Rating:** 5
**Confidence:** 3

**Summary:**

Constrained Markov decision processes (CMDPs) are environments subjected to constraints and have been addressed extensively using reinforcement learning techniques. Few sample-based planning algorithms have been applied to such problems. The paper introduces an MCTS-based approach for learning cost-conscious policies for CMDPs.

**Strengths:**

**Orignality:** While not introducing a completely novel approach, they apply an offline learning technique to estimate costs in CMDPs online.

**Significance:** The contributions of the paper lack significance.

**Clarity:** The paper is understandable.

**Weaknesses:**

The main drawback of the paper is its lack of significance. The approach introduced is not novel. Learning values/cost estimates offline to be applied online is not a new idea. Nor is the learning approach using a novel technique.

I also question the soundness of the analysis. In Prop. 1, the authors claim that at each iteration of their algorithm, they are guaranteed to find the optimal solution. They base their claim on the proof in [Kocsis & Szepesvari, 2006]. However, that work states that UCT is guaranteed to converge in the limit --- it is not guaranteed to converge in finite-time.

The experiments also do not demonstrate the claimed performance improvements achieved by their approach. While C-MCTS significantly outperforms CC-MCP in Rocksample, it does not necessarily outperform MCTS. Nor does it significantly have less cost violations than CC-MCP. Furthermore, CC-MCP searches deeper than C-MCTS in Rocksample(11, 11).

The experiments themselves also do not seem. Maybe I'm missing something but it seems, that C-MCTS gets extra computations. The cost for pre-training should be factored into the planning budget.

**Questions:**

Corollary 1: How can you guarantee optimality if you prune sub-trees for which the cost is over-estimated?

What exactly is a "boundary region?"

The lengths of the arrows are difficult to distinguish in Fig. 4. I would suggest finding a different way to visualize the number of times an action is selected.

---

> ### Author Response · Authors · 2023-11-16
>
> First and foremost we want to thank the reviewer for the valuable feedback.
>
> > The main drawback of the paper is its lack of significance. The approach introduced is not novel. Learning values/cost estimates offline to be applied online is not a new idea. Nor is the learning approach using a novel technique.
>
> We agree with the reviewer that learning offline (e.g. pretraining) and using this knowledge online is by itself not a new idea; also, safety critics have been around for some time and extensively being used in contemporary research. However, we want to emphasize that it is not straightforward to use cost estimates learned in a pre-training stage in combinations with planning methods such as MCTS as care must be paid to policies being different/emerging over time while safety critics usually approximate a cost function only for a fixed policy. This is why we came up with a doubly-looped iteration cycle to collect training data while optimizing the Lagrange multiplier to solve an MDP without violating constraints (in final deployment).
>
> > I also question the soundness of the analysis. In Prop. 1, the authors claim that at each iteration of their algorithm, they are guaranteed to find the optimal solution. They base their claim on the proof in [Kocsis & Szepesvari, 2006]. However, that work states that UCT is guaranteed to converge in the limit --- it is not guaranteed to converge in finite-time.
>
> Fair point. As MCTS with upper confidence bounds converges asymptotically to the optimal policy [1], usually a time- or computational budget-limit is used to terminate learning [2,3]. In our case, since we are interested in a feasible solution, we terminate the training process (search for $\lambda^*$ effectively) in the data gathering phase only when enough data has been gathered and the cost constraints are satisfied (see ``data gathering'' phase in Fig. 1).
>
> To solve each of the MDPs we use a sufficiently high number of simulations that ensures a nearly optimal policy (Refer Figure 5, comparing performance for low and high number of simulations). When the costs observed are close to the constraint (but not violated), determined by $\hat{c}-\epsilon \leq V_{C}^{k, *} \leq \hat{c}$,  the iterative process is terminated.
>
> We have adapted the text accordingly to clarify this.
>
> [1] Kocsis, Levente, and Csaba Szepesvári. "Bandit based monte-carlo planning." European conference on machine learning. Berlin, Heidelberg: Springer Berlin Heidelberg, 2006.
>
> [2] Silver, David, et al. "Mastering the game of Go with deep neural networks and tree search." nature 529.7587 (2016): 484-489.
>
> [3]  Schrittwieser, Julian, et al. "Mastering atari, go, chess and shogi by planning with a learned model." Nature 588.7839 (2020): 604-609.
>
> > The experiments also do not demonstrate the claimed performance improvements achieved by their approach. While C-MCTS significantly outperforms CC-MCP in Rocksample, it does not necessarily outperform MCTS. Nor does it significantly have less cost violations than CC-MCP. Furthermore, CC-MCP searches deeper than C-MCTS in Rocksample(11, 11).
>
> The experiments were evaluated with each of the three algorithms being run 100 times (or episodes).  While focusing solely on average reward and cost values (top, middle rows in Fig 2.) might suggest MCTS performs best, closer inspection reveals a drawback. MCTS adheres to cost constraints only on average. The bottom row plots (in Fig 2.) indicate a high percentage of episodes with cost violations, implying that across 100 episodes, many violate cost, with some being highly conservative. Hence the cost constraint is fulfilled on average but with a high probability of violating constraints. C-MCTS on the other hand fulfills the cost constraint with greater consistency and hence is more favorable in practice.
>
> Only in Rocksample(11, 11) CC-MCP searches deeper, but this results in a very poor performance. The search tree is probably extending towards a non-optimal solution space.
>
> > The experiments themselves also do not seem. Maybe I'm missing something but it seems, that C-MCTS gets extra computations. The cost for pre-training should be factored into the planning budget.
>
> It is correct that C-MCTS requires additional costs for pre-training. The most expensive part of this phase is generating training data. However, this could be optimized by using offline data or re-using simulated data. Our focus was not on optimizing the training phase but on showcasing its benefits during deployment. Nevertheless, the pre-training costs could still be accrued over the lifetime of an agent when:
> - C-MCTS requires fewer planning iterations during deployment.
> - C-MCTS can use approximate planning models (not reliant on the planner for cost estimates).

---

> ### Author Response · Authors · 2023-11-16
>
> > Corollary 1: How can you guarantee optimality if you prune sub-trees for which the cost is over-estimated?
>
> We cannot guarantee optimality, only a feasible solution, as written in the Corollary and in the Proof sketch – this is what we meant by “This leads to a safe, but potentially conservative behavior.” To communicate this more clearly, we have rephrased to the following in the proof sketch paragraph:
>
> “As discussed before, the inner training loop will always converge to the optimal solution for the k-th MDP. In case the safety critic over-estimates the expected cost, it prunes the corresponding branch in the MCTS tree. This leads to a safe, but potentially conservative (i.e., non-optimal) behavior”
>
> > What exactly is a "boundary region?"
>
> We acknowledge that this phrasing is not very clear. What we mean there is that elaborating further from Definition 3 on cost-non-critical states, there exist a hyper-surface (similar to the decision boundary of a support vector machines classifier) that contains the cost-critical states, i.e. the states where selecting a specific action (over others) has a high effect on expected feature performance.
>
> We have edited the related part in the beginning of Section 3.1.1 to clarify this point.
>
> > The lengths of the arrows are difficult to distinguish in Fig. 4. I would suggest finding a different way to visualize the number of times an action is selected.
>
> Thank you for your comment. We share your opinion and for sake of clarity we would just remove the arrows as they do not contribute much to convey the message this figure is supposed to. Would the reviewer agree?

---

> > ### Comment · Reviewer_EfYF · 2023-11-22
> >
> > I appreciate the authors' feedback. However, while my score is slightly increased, I just do not think the evaluation is strong enough and I'm still a iffy on the theoretical guarantees.

---

### Official Review · Reviewer_q5zX · 2023-11-03

**Soundness:** 3 good
**Presentation:** 3 good
**Contribution:** 2 fair
**Rating:** 5
**Confidence:** 3

**Summary:**

This paper proposes a novel algorithm, C-MCTS, for solving problems with constraints.
The proposed algorithm is tested on two small benchmark problems and shows improving performances compared to the baseline algorithms, CC-MCP and vanilla MCTS.

**Strengths:**

- C-MCTS achieves improving performance in the quality of the solutions while not violating the cost constraints.

**Weaknesses:**

- The actual running time of the experiments needs to be provided.

- Several points need to be clarified in the explanation that is described in the Questions below.

**Questions:**

- About proof of Proposition 1 in P.4.
"... guaranteed to find the optimal solution in the k-th MDP."
UCT's proof is asymptotic, and we need an infinite number of simulations to guarantee the convergence to the optimal.
Also, we have no way of knowing whether we actually reached the optimal (in my understanding).
Does this proof assume that we do an infinite number of simulations on each iteration? Is that valid?

- Also, I may have overlooked, but how long does this step take in the actual experiments in Section 4? (is it the "Solve MDP" step in Fig. 1?)

- How meaningful is it to count the number of cost violations?
In what problem settings are we not allowed to violate the cost even during the simulations?

- For Fig. 2, how was the actual execution time of the three algorithms?
Also, for the Safe GridWorld experiment, did C-MCTS with $2^9$ planning iterations and CC-MCP with $2^{20}$ planning iterations take a similar amount of time?

- About Fig. 4.
What is one episode here?
Does one episode for C-MCTS and CC-MCP require a similar amount of time?
If the latter is faster, could we run CC-MCP 100 times and select the path based on the majority?

Minor question.
- I got confused with the notation of $C, c, \hat{c}$. Are these sets or functions?

---

> ### Author Response · Authors · 2023-11-16
>
> We thank the reviewer for the thorough review and time and effort spent on our paper. We are happy that our contribution that C-MCTS improves the performance (in terms of reward) while not violating cost constraints is recognized.
>
> > About Proposition 1 in P.4. "... guaranteed to find the optimal solution in the k-th MDP." UCT's proof is asymptotic, and we need an infinite number of simulations to guarantee the convergence to the optimal. Also, we have no way of knowing whether we reached the optimal (in my understanding). Does this proof assume that we do an infinite number of simulations on each iteration? Is that valid?
>
> Fair point. As MCTS with upper confidence bounds converges asymptotically to the optimal policy [1], usually a time- or computational budget-limit is used to terminate learning [2,3]. In our case, since we are interested in a feasible solution, we terminate the training process (search for $\lambda^*$ effectively) in the data gathering phase only when enough data has been gathered and the cost constraints are satisfied (see ``data gathering'' phase in Fig. 1).
>
> To solve each of the MDPs we use a sufficiently high number of simulations that ensures a nearly-optimal policy (refer Figure 5, comparing performance for low and high number of simulations). When the costs observed are close to the constraint (but not violated), determined by $\hat{c}-\epsilon \leq V_{C}^{k, *} \leq \hat{c}$,  the iterative process is terminated.
>
> We have adapted the text accordingly to clarify this.
>
> [1] Kocsis, Levente, and Csaba Szepesvári. "Bandit based monte-carlo planning." European conference on machine learning. Berlin, Heidelberg: Springer Berlin Heidelberg, 2006.
>
> [2] Silver, David, et al. "Mastering the game of Go with deep neural networks and tree search." nature 529.7587 (2016): 484-489.
>
> [3]  Schrittwieser, Julian, et al. "Mastering atari, go, chess and shogi by planning with a learned model." Nature 588.7839 (2020): 604-609.
>
> > Also, I may have overlooked, but how long does this step take in the actual experiments in Section 4? (is it the "Solve MDP" step in Fig. 1?)
>
> To solve the k-th MDP during the iterative training process, we use 1024 simulations for Rocksample(5,7) and Rocksample(7,8), and 512 simulations for Rocksample(11,11) and Safe Gridworld (Refer to column 'Planning Iterations' in Table 2 in A.2).
>
> > How meaningful is it to count the number of cost violations? In what problem settings are we not allowed to violate the cost even during the simulations?
>
> Let us answer depending on the actual question you ask:
> - If the question is related to the type of constraints selected for the experimental evaluation, then we would like to clarify that the safety critic can be trained to approximate any type of safety or cost constraint, like number of cost violations, magnitude of violation or combinations of them. We selected environments that count the number of constraint violations to enable a more clear evaluation of the results and comparison with the baseline.
> - If the question is related to how we evaluate C-MCTS, e.g. in Figs. 2 (bottom row) and 4 (bottom row): please note that here we count the cost constraint violations during /actual/ deployment of the agent (of course we still run the agent in a simulator, but here we do not count the violations that the agent would do in the simulation step of MCTS itself). Hence, this would lead to an actual possibly harmful execution of an action in the real world. Here we also stick to the convention that constraints should be hard and not being violated.

---

> ### Author Response · Authors · 2023-11-16
>
> > For Fig. 2, how was the actual execution time of the three algorithms? Also, for the Safe GridWorld experiment, did C-MCTS with planning iterations and CC-MCP with planning iterations take a similar amount of time?
>
> In terms of computational cost per simulation across the different algorithmic phases:
> - Selection: CC-MCP is the most computationally expensive, requiring more operations to select the best child node. MCTS and C-MCTS have identical operation counts.
> - Expansion: C-MCTS incurs additional costs due to the safety critic's prediction. MCTS and CC-MCP require no additional computation during expansion.
> - Backpropagation: CC-MCP backs up Q-values for both reward and cost, while MCTS and C-MCTS back up only Q-values for reward.
> - Rollout: Computational cost is identical for C-MCTS, MCTS, and CC-MCP.
> C-MCTS and MCTS algorithms were implemented in Python, while the benchmark CC-MCP uses a C++ implementation. Comparing actual execution times was unfair since our implementation was not optimized for hardware efficiency. Also, such an optimization would highly depend on the hardware platform (e.g. CPUs vs GPUs). For instance, added cost in C-MCTS's expansion phase is highly parallelizable, with good potential for effective GPU utilization. So, for our analysis we instead compare the number of simulations (planning iterations) required by each algorithm as a performance metric.
>
> We have also added this information to the appendix of the paper.
>
> > About Fig. 4. What is one episode here? Does one episode for C-MCTS and CC-MCP require a similar amount of time? If the latter is faster, could we run CC-MCP 100 times and select the path based on the majority?
>
> Here, one episode corresponds to an agent starting from one of the four squares in the bottom-left of the grid and reaching the green square in the top-right corner. Fig 4. was evaluated for 100 episodes i.e., both CC-MCP and C-MCTS were run 100 times each. C-MCTS uses $2^9$ planning iterations and CC-MCP uses $2^{20}$ planning iterations, hence the former is much cheaper. Here, running CC-MCP more times wouldn’t improve its performance since the cost estimates for CC-MCP are dependent on the planning model. A model mismatch in the planner has introduced inaccuracies, rendering additional runs less beneficial.
>
> > I got confused with the notation of $C$, $\hat{C}$. Are these sets or functions?
>
> We apologize for not being clear at this point. Following the notation convention of [1], $\mathbf{C}=\\{C_m\\}\_\{1\\ldots M\}$ is a set of $M$ non-negative cost functions, with $\mathbf{\hat{c}} = \{c_m\}_{1\ldots M} \in [0,1]$ their respective thresholds. In the text, we assume only one constraint function $C$ with its respective threshold $\hat{c}$, to simplify the notation.
>
> In any case, we have re-written this point more clearly in the new version of the text.
>
> [1] Lee, Jongmin, et al. "Monte-Carlo tree search for constrained POMDPs." Advances in Neural Information Processing Systems 31 (2018).
>
> We hope that we answered your questions adequately and eliminated your concerns. We are happy to discuss further if any additional points come up or if things are not yet clear.

---

> > ### Comment · Reviewer_q5zX · 2023-11-22
> > **Reply to the Authors' comments**
> >
> > I thank the authors for their detailed replies. However, I find them insufficient to change my evaluation.

---

### Meta-Review · Area_Chair_BHR6 · 2023-12-14

**Metareview:**

The paper introduces a variant of MCTS for safety-critical decision-making tasks subject to constraints using the formulation of Constrained Markov Decision Process (CMDP) formulation. With the advantage that C-MCTS operates closer to the constraint boundary while satisfying cost constraints, the algorithm may achieve incremental improvements.

There is a lack of simulation experiments, and comparisons with benchmark CMDP algorithms. While the paper discusses the potential for real-world applications, the lack of empirical validation limits the practical relevance of the research.

**Justification For Why Not Higher Score:**

The meta reviewer read the paper and discussions and agrees with all the reviewers that this paper hasn't reached the bar of ICLR.

**Justification For Why Not Lower Score:**

N/A

---

### Decision · Program_Chairs · 2024-01-16

Reject